# Development of Ultrafiltration Kaolin Membranes over Sand and Zeolite Supports for the Treatment of Electroplating Wastewater

**DOI:** 10.3390/membranes12111066

**Published:** 2022-10-29

**Authors:** Hajer Aloulou, Wala Aloulou, Joelle Duplay, Lassaad Baklouti, Lasâad Dammak, Raja Ben Amar

**Affiliations:** 1Research Unit Advanced Technologies for Environment and Smart Cities, Faculty of Sciences, University of Sfax, UR22ES02, BP1171, Sfax 3000, Tunisia; 2ITES-Institut Terre et Environnement de Strasbourg, Université de Strasbourg, UMR 7063 CNRS, 5, Rue René Descartes, 67084 Strasbourg, France; 3Department of Chemistry, College of Sciences and Arts at Ar Rass, Qassim University, Ar Rass 51921, Saudi Arabia; 4Université Paris-Est Créteil, CNRS, ICMPE, UMR 7182, 2 Rue Henri Dunant, 94320 Thiais, France

**Keywords:** kaolin, ceramic composite membrane, ultrafiltration, oily wastewater, heavy metals

## Abstract

A high cost of high-purity materials is one of the major factors that limit the application of ceramic membranes. Consequently, the focus was shifted to using natural and abundant low-cost materials such as zeolite, clay, sand, etc. as alternatives to well-known pure metallic oxides, such as alumina, silica, zirconia and titania, which are usually used for ceramic membrane fabrication. As a contribution to this area, the development and characterization of new low-cost ultrafiltration (UF) membranes made from natural Tunisian kaolin are presented in this work. The asymmetric ceramic membranes were developed via layer-by-layer and slip-casting methods by direct coating on tubular supports previously prepared from sand and zeolite via the extrusion process. Referring to the results, it was found that the UF kaolin top layer is homogenous and exhibits good adhesion to different supports. In addition, the kaolin/sand and kaolin/zeolite membranes present an average pore diameter in the range of 4–17 nm and 28 nm, and water permeability of 491 L/h·m^2^·bar and 182 L/h·m^2^·bar, respectively. Both membranes were evaluated in their treatment of electroplating wastewater. This was done by removing oil and heavy metals using a homemade crossflow UF pilot plant operated at a temperature of 60 °C to reduce the viscosity of the effluent, and the transmembrane pressure (TMP) of 1 and 3 bar for kaolin/sand and kaolin/zeolite, respectively. Under these conditions, our membranes exhibit high permeability in the range of 306–336 L/h·m^2^·bar, an almost total oil and lead retention, a retention up to 96% for chemical oxygen demand (COD), 96% for copper and 94% for zinc. The overall data suggest that the developed kaolin membranes have the potential for remediation of oily industrial effluents contaminated by oil and heavy metals.

## 1. Introduction

Among different alternative technologies, membrane separation using porous ceramic membranes appears to be promising and efficient in environmental applications, especially in industrial wastewater treatment [1,2,3,4,5]. Recently, ceramic membranes have attracted the attention of researchers and industrialists due to their excellent separation efficiency, high chemical and thermal stabilities towards the harsh environment, good mechanical resistance to high-pressure conditions, long durability, relatively low fabrication cost, and reduced tendency to fouling [6,7,8,9]. Several studies proved that the fabrication of ceramic membranes from commonly inorganic available materials such as alumina, titania, silica and zirconia [10,11,12] is very expensive and requires high sintering temperature (>1300 °C) [13]. From the economic point of view, the membranes derived from these costly raw materials are not suitable for industrial applications. For this reason, many researchers have focused on developing low-cost ceramic membranes using natural raw materials such as clay, sand, zeolite, sepiolite, dolomite, perlite, phosphate, etc. due to their abundance and lower sintering temperatures compared to those needed for metal oxide materials [14,15,16,17,18,19,20]. Of these natural materials, natural kaolin clay was utilized as the main raw material for preparing cheaper membranes [21]. It offers excellent strength, low plasticity and good hydrophilicity to the membranes [22]. In addition, kaolin presents good mineralogical properties and an adequate chemical composition as well as an appropriate particle size for the development of good ceramic porous membranes [23].

The porous ceramic membranes are commonly asymmetrical in their microstructures and are constituted from several layers using the same or different ceramic materials. They generally consist of a thin layer which assures selectivity and a macroporous support. In some cases, intermediate layers are also included to adapt the overall pore size. Typically, a good ceramic support provides a sufficient mechanical strength and high porosity [24]. The top layer is the selective layer, which provides the membrane separation process, while the intermediate layers bridge between the support and the top layer and gradually reduce the pore size [25]. 

Low-cost ceramic membranes need low-cost raw material composition, an easy manufacturing process usually used by the traditional ceramic industry (pressing or extrusion) and lower sintering temperatures than those used in commercial ceramic membranes, all of which reduces processing costs [25]. The approximate prices of commercial ceramic membranes based on alumina and zirconia are between 500 and 3000 $·m^−2^ [26], offering low-cost ceramic membranes a wide margin for commercialization. Generally, the price of alumina membrane is approximately 100 times higher than that of kaolin membrane. This difference in price is mainly caused by the raw materials and the different manufacturing conditions, particularly during sintering. The energy necessary for the sintering of alumina membrane is much higher than that of a kaolin one, due to the large difference in sintering temperature (approximately 1600 °C for alumina and 1000 °C for kaolin membrane). Additionally, kaolin membranes display a lower apparent density (2.7 g·cm^−3^) than alumina membranes (3.98 g·cm^−3^); consequently, the amount of material needed per unit area decreases and the cost of raw materials is reduced [27].

Large volumes of oily wastewater discharged from oil exploration, storage and transportation industries, refineries, and from other industrial activities are considered to be one of the major contaminants to the environment and threats to human health [28,29,30]. More particularly, oily wastewater, which is an extremely hazardous pollutant worldwide, is generated from electroplating industry and is usually composed of organic materials and heavy metals [31,32]. Efficient treatment of these pollutants is necessary both from environmental and human health aspects. Although free-floating oils (superior to 150 μm), unstable dispersed oil (between 20 and 150 μm) [33] and different metal ions such as Zn, Cu, Hg, Ni, Cd, Pb, and Cr can be removed by conventional techniques such as adsorption [34,35,36] and coagulation–flocculation [37,38]. However, for the effective removal of emulsified oils (d < 5 μm) and heavy metals, the utilization of more successful methods, such as membrane filtration, is required [39,40].

The objective of this work is the development and the characterization of new composite UF membranes based on sand and zeolite supports via layer-by-layer and slip-casting methods. The performances of kaolin/sand and kaolin/zeolite membranes were investigated through the treatment of industrial oily wastewater contaminated with heavy metals and oil. Several experimental parameters were determined to evaluate the UF performances such as permeate flux, removal of turbidity, COD (Chemical Oxygen Demand), oil, and heavy metals. The membrane fouling and cost were estimated to confirm the efficiency of the prepared membrane in the industrial applications.

## 2. Experiment

### 2.1. Materials

Natural kaolin clay, previously utilized in our laboratory for the fabrication of a monolayer UF membrane [4], was chosen as the raw material for top-layer preparation. The clay powder, collected from the Tabarka region (northwest Tunisia), was principally composed of a large quantity of silica SiO_2_ (55.25%) and alumina Al_2_O_3_ (24.17%). Moreover, the mineralogical analysis showed the presence of 61% of kaolinite and 39% of illite, while the non-clayey minerals were essentially represented by quartz [4].

Two tubular supports from natural sand and zeolite were used for the preparation of new composite membranes: kaolin/sand and kaolin/zeolite [14,15]. The characteristics of the different supports are summarized in Table 1.

Polyvinyl alcohol PVA (Rhodoviol 25/140), used as an additive, was purchased from Prolabo and ethanol as a rinsing solvent from Chemi-Pharma (Cebalat, Tunisia).

### 2.2. Fabrication and Characterization of Kaolin Membranes 

The sand and zeolite supports were rinsed first with hot water, and then with ethanol via ultrasound vibration to remove residual particles. After that, the cleaned supports were dried overnight at 100 °C. For the coating layer, two suspensions were prepared differently by the composition, previously optimized for the coating of sand and zeolite supports [1,15]:8% of kaolin powder (ϕ < 53 µm) was mixed with 62% of water and 30% of PVA (12 wt% aqueous solution) for kaolin/sand membrane.2% of kaolin powder (ϕ < 53 µm) was mixed with 68% of water and 30% of PVA (12 wt% aqueous solution) for kaolin/zeolite membrane.

Then, the sand and zeolite supports were coated, respectively, via layer-by-layer and slip-casting methods, as described elsewhere [14,41]. Finally, the green membranes were kept in the air for 24 h, then were sintered in a programmable muffle furnace before characterization and application. Thermal cycling was performed in two steps: the first annealing at 250 °C for 3 h to remove residual water and organic additives, and the next annealing at 900 °C for kaolin/sand and 850 °C for kaolin/zeolite for 3 h to ensure the sintering of the membranes.

The membrane morphology was observed using a scanning electron microscope (SEM) (MERLIN scanning electron microscope by ZEISS associated with a GEMINI II column, Göttingen, Germany). The surface elemental composition of the samples was also analyzed using energy dispersive X-ray (EDX) fitted to the SEM equipment. The average pore sizes of the membranes were estimated via the BJH (Barret-Joyner-Halenda) model [42]. Water permeability of the different membranes was assessed using a stainless-steel unit and calculated following Darcy’s law [4,15].

### 2.3. Ultrafiltration Experiment

The efficiency of kaolin membranes, kaolin/sand, and kaolin/zeolite was evaluated by filtration of wastewater coming from an electroplating industrial plant (SOPAL) located in Sfax, Tunisia during 1 h at ambient temperature under transmembrane pressure of 1 bar and 3 bar, respectively. In fact, the oily wastewater contaminated with heavy metals (Pb, Zn and Cu) was unsuitable for direct use as feed for UF, because it contains free-floating oil on the top and solid particles at the bottom, which can cause the blockage of the membrane pores. As a pretreatment, the effluent was filtrated by a sieve of 100 μm before the UF experiment to remove large particles. 

The characterization of raw and treated wastewater was determined by measuring the pH, the conductivity, the turbidity, the COD, the oil, and heavy metal content. The turbidity was measured by a turbidimeter (model 2100A, Hach, USA) in agreement with standard method 2130B. The COD was obtained using a colorimetric technique (COD 10119, Fisher Bioblock Scientific, Illkirch, France). The conductivity and pH were measured by a conductimeter (EC-400L, Istek, Seoul, Korea) and a pHmeter (pH-220L, Istek, Seoul, Korea). The content of the oil and heavy metals retention was measured by determining the concentration in the feed and solutions using a UV-spectrophotometer (UV-9200) at a wavelength of 363 nm and atomic absorption spectroscopy (AAS), respectively. For the evaluation of UF rejection, the rejection of the different parameters (COD, turbidity, oil and heavy metals) was determined by Equation1 [43]: (1)$$R_{X}=\frac{\left(X_{feed}-X_{permeate}\right)}{X_{feed}}\times 100$$
where *X_feed_* and *X_permeate_* are the values of turbidity, and COD represents the values of raw and treated wastewater, respectively.

The calibration curve is used to calculate the oil concentration (C) in industrial wastewater. The following steps are followed:−Determination of λ_max_.−Measurement of absorbance A for each standard solution and for the dosed solution.−The calibration curve is plotted for the standard solutions: A = f(C).−The absorbance A of the dosed solution is plotted on the calibration curve to determine its concentration.

The viscosity of the raw effluent at 25 °C and 60 °C was measured by a rotary viscosimeter Tve-05 (LAMY) because 60 °C was selected as a suitable temperature for the purification of this type of oily wastewater in a previous work [43].

### 2.4. Modeling of Membrane Fouling

To describe the decrease in the membrane flux during the treatment of the electroplating wastewater by the ultrafiltration process, the model of Hermia based on four empirical approaches, namely complete pore blocking (Equation (2)), standard pore blocking (Equation (3)), intermediate pore blocking (Equation (4)), and cake filtration (Equation (5)), could be widely employed [5]:(2)$$Ln\left(J^{-1}\right)=Ln\left(J_{0}^{-1}\right)+K_{b}t$$
(3)$$J^{-0.5}=J_{0}^{-0.5}+K_{s}t$$
(4)$$J^{-1}=J_{0}^{-1}+K_{i}t$$
(5)$$J^{-2}=J_{0}^{-2}+K_{c}t$$
where *J* is the permeate flux, *t* is the time of filtration, *J*_0_ is the y-intercept, and *K* is the slope.

### 2.5. Determination of Fouling Resistances and Membrane Regeneration 

The antifouling characteristics of the prepared membranes kaolin/sand and kaolin/zeolite were evaluated under optimal pressures of 1 and 3 bar for one hour. Several antifouling parameters, namely Flux Decay Ratio (FDR) and Flux Recovery Ratio (FRR), could be calculated using the following equations [44]:(6)$$FDR=\frac{J_{w}-J_{c}\;}{J_{w}}\times 100$$
(7)$$FRR=\frac{J_{wa}\;}{J_{w}}\times 100$$

*J_w_* is the water flux of the new membrane and *J_c_* is the stabilized permeate flux during the UF of the oily wastewater. *J_w__a_* is the water permeate flux of the membrane measured after rinsing the used membrane with distilled water.

The membrane regeneration was accomplished initially by water rinsing followed by an acid–basic treatment with an alternative circulation of 2% solutions of NaOH at 80 °C and HNO_3_ at 60 °C for 30 min. At the end, the membrane was rinsed with distilled water until neutral pH was obtained. The efficiency of the cleaning protocol was confirmed by measuring the water permeability after the cleaning cycle, which must be almost equal to that of the new membrane.

## 3. Results and Discussion

### 3.1. Characterization of Kaolin Membranes 

#### 3.1.1. Membrane Morphology 

The microstructure of the top surface of kaolin membranes is shown in Figure 1 and Figure 2. Figure 1a–c depict the surface morphology of kaolin/sand membrane sintered at 800 °C, 900 °C and 1000 °C, respectively. It is clear from SEM micrographs that for the two membranes, the surface is homogenous and without cracks. In addition, the kaolin particles are uniformly distributed on the porous ceramic sand support. 

At 800 °C and 900 °C, grains partially join together to obtain a stronger ceramic body, and the membranes present an appropriate porous structure [3]. For the temperature of 1000 °C, the densification process starts, and particles are closer [45]. The membrane then loses its microporous structure. Therefore, either 800 °C or 900 °C can be chosen as optimal sintering temperatures for the new composite membrane kaolin/sand, but we will select 900 °C because the kaolin particles appear denser at the membrane surface and the pores are more closed.

Figure 2 presents a good deposition of the kaolin particles on the surface of the kaolin/zeolite membranes sintered at different temperatures. Furthermore, the membrane surface exhibits a homogeneous morphology and good adhesion. Figure 2a shows that the membrane sintered at 820 °C has a smoother surface, with the presence of large and non-uniform pores that will affect the membrane filtration performance. By increasing the temperature to 850 °C (Figure 2b) and 880 °C (Figure 2c), the intergranular contact between particles was reduced and small pores were created. So, to minimize energy, 850 °C was selected as an optimal sintering temperature for the kaolin/zeolite membrane.

EDX spectra analysis of kaolin/sand and kaolin/zeolite membranes (Figure 3a–b) clearly shows relative signals of Si, Al, and O, most characteristic elements of clay, as well as the presence of K, Ca, Mg, and Na chemical elements in the composition of the used kaolin clay [4]. The distribution of these elements confirms the success of the coating of top layers on sand and zeolite supports. The small peak for carbon could be due to impurities due to improper handling of samples without wearing gloves.

#### 3.1.2. Determination of Pore Diameters

From Figure 4, it can be concluded that the pore diameters for the kaolin composite membranes are in the range of 4–17 nm for kaolin/sand and 26–34 nm for kaolin/zeolite. Meanwhile, in the works of Aloulou et al. [14,15], the preparation of sand and zeolite supports showed pore sizes of 10.36 µm and 0.55 µm, respectively. This result indicates an important reduction of the pore size, especially for kaolin/sand membrane. These latter composite membranes are classified as mesoporous and could be good candidates to be applied in the ultrafiltration domain.

#### 3.1.3. Determination of Water Permeability

Steady-state water permeates flux at different pressures, from 0 to 1 bar for kaolin/sand membrane and from 0 to 5 bar for kaolin/zeolite membrane, are illustrated in Figure 5. It was established that the relationship between permeate flux and transmembrane pressure is linear, and that therefore the permeability of the membrane could be calculated by determining the slope of the curve. It is found that water permeability of the new composite membranes is 491 L/h·m^2^·bar for kaolin/sand and 182 L/h·m^2^·bar for kaolin/zeolite.

Indeed, the sand support is very porous, presenting pore diameters of 10.36 µm and a water permeability of 3611 L/h·m^2^, which explains the presence of the flux at 0 bar. This behavior is identical to that of the sand MF membrane [15].

### 3.2. Application to the Treatment of Electroplating Wastewater

The performance of kaolin composite membranes (kaolin/sand and kaolin/zeolite) was assessed by the treatment of the oily wastewater contaminated with heavy metals. The physicochemical parameters of the raw wastewater are reported in Table 2. It was observed that the viscosity of the effluent decreases from 34.4 × 10^–3^ Pa·s at 25 °C to 1.3 × 10^–3^ Pa·s at 60 °C which represents the producing effluent temperature.

#### 3.2.1. Fluxes and Rejection Efficiencies 

Membrane filtration experiments were carried out using the prepared kaolin membranes at 60 °C, shown also to be the optimal temperature for the treatment of oily wastewater [43], and a transmembrane pressure of 1 bar for kaolin/sand and 3 bar for kaolin/zeolite. Figure 6 presents the variation in the permeate flux (J) of the wastewater with filtration time. For kaolin/sand membrane, a quick and significant flux reduction was observed during the first 20 min from 579 to 359 L/h·m^2^, resulting in a stabilized permeate flux (J_c_) of 336 L/h·m^2^ obtained after 30 min. Nevertheless, for kaolin/zeolite membrane, a much lower flux reduction was observed from 345 to 315 L/h·m^2^ during the first 10 min, then the permeate flux was stabilized at 306 L/h·m^2^ after only 15 min. Therefore, it seems that the fouling is more important for the kaolin/sand membrane. This behavior was usually observed with membranes coated in the sand support, which had excellent porosity (44.72%), large pore size (10.36 µm) and high-water permeability (3611 L/h·m^2^·bar) [2,15]. This behavior can be explained by the formation of a cake layer during the retention of the pollutants, which constitute a dynamic layer superimposed onto the initial membrane surface.

Rejection efficiency of the UF kaolin membranes was determined by measuring the rejection rate of the different pollutants. Figure 7 shows an almost total retention of oil and turbidity, as well as a COD removal of 98% for both membranes. In addition, it seems that the kaolin membranes displayed simultaneous and encouraging abilities to remove high metals such as Pb (99%), Cu (>96%) and Zn (94%) at transmembrane pressure of 1 bar for kaolin/sand and 3 bar for kaolin/zeolite. This result is similar to that obtained by Aloulou et al. [43], who used a composite UF membrane made from a mixture of commercial nanoparticles TiO_2_ and synthetic smectite nanocomposites over a zeolite support, which highlighted the prominence of these new natural kaolin-based membranes.

Generally, it is well known that the UF cannot be applied directly to remove heavy metals from wastewater due to its comparatively high pore size (2–50 nm). However, the separation of metal ions by the charged UF membranes can be explained by ion repulsion. This method attempts to offer excellent heavy metal removal efficiency with a high permeate flux [46]. Based on the literature, Yao et al. [47] observed a total chromium (Cr^VI^) rejection by the new positively charged UF membranes. In our case, we obtain high heavy metals rejection (>94%) for Pb^2+^, Cu^2+^ and Zn^2+^ using kaolin UF membranes, knowing that kaolinite particles exhibit sites with an amphoteric character and present a surface positive charge [48].

#### 3.2.2. Fouling Mechanisms

Permeate flux of kaolin/sand sharply decreases during the first 20 min due to colloidal and suspended particles being presented in the feed, which both contributed to pore blocking and denoted fast clogging during the rejection by the membrane [49]. Then, it slowly and continuously decreases during the filtration due to slow pore clogging [18]. Whereas the kaolin/zeolite membrane presents a very slight decrease in the permeate flux during the first minutes of filtration and then stabilizes after 15 min. This behavior can be explained by an establishment of an instantly fouling layer on the membrane surface.

Figure 8a,d show the linearized plots of pore blocking models using kaolin membranes, and Table 3 illustrates the associated parameters to the considered models in terms of slope, y-intercept and R^2^. It is evident that the model describing experimental data with the best R^2^ value (Almost 1) is considered to refer the suitable fouling mechanism [50]. According to R^2^ values, it seems that the cake filtration model can describe the fouling for kaolin/sand membrane (Figure 8d). This behavior can be explained by the deposition of particles larger than the membrane pore size onto the membrane surface. For kaolin/zeolite membrane, R^2^ is relatively low (<0.900); therefore, it can be deduced that the Hermia model did not correlate with the experimental data.

#### 3.2.3. Antifouling and Cleaning Study

The fouling is coupled with an evident deterioration of the membrane surface during the ultrafiltration process. To explain the fouling resistance ability of the kaolin membranes, Flux Recovery Ratios (FRR) and Flux Decay Ratio (FDR) were calculated and presented in Figure 9.

In fact, FRR values indicate the percentage recovery of the original water permeability of the membrane after filtration test and rinsing with water; besides, FDR values show the percentage of the flux decays during the filtration process.

From Figure 9, the kaolin/sand membrane presents the highest FRR value (55.8%) and the lowest FDR value (31.5%). It is worth noticing that higher FRR values and lower FDR values are more beneficial and prove better antifouling properties [51]. Consequently, the utilization of sand support for the kaolin membrane enhanced the permeate flux and the antifouling properties. In addition, as depicted in Figure 9, the permeate flux of the developed membranes recovers to 55.8% of the initial flux (FRR = 55.8%) for kaolin/sand and only 37.68% (FRR = 37.68%) for kaolin/zeolite. Therefore, for the oily wastewater ultrafiltration, the results confirm an intensive membrane fouling (>26%) requiring chemical cleaning to recover the initial kaolin membranes performances [52]. 

The efficiency of the membrane regeneration was determined by checking water permeability. Figure 10a,b represent the evolution of the water permeation flux, with the transmembrane pressure for the new and the regenerated kaolin membranes shown. The results demonstrate that the water permeability values were very close, confirming the efficiency of the cleaning process used.

### 3.3. Membranes’ Cost Estimation

The industrial aspect of membrane technology is linked to its cost which should be competitive. Ceramic membrane is well known for its high cost in comparison to polymeric membrane. Membrane cost is always estimated from the cost of the raw material and energy consumption. In general, the common ceramic membranes are made of metallic oxide such as alumina and zirconia. Therefore, they are more expensive than those using natural and abundant raw materials such as clay, zeolite, and sand, because of the costly synthetic materials used. The cost of the commercial membrane made of α-Al_2_O_3_ is reported to be 989–1220 ($/m^2^) [53]. Table 4 shows details related to the cost estimation of kaolin membranes (150 mm of length, 5 mm of inner diameter) including the price of raw materials used and the energy needed for shaping and sintering. The fabrication cost of kaolin membranes was estimated as 17.88 $/m^2^ for kaolin/sand and 20.92 $/m^2^ for kaolin/zeolite. These values remain relatively lower than those reported by some researchers for low-cost-developed ceramic membranes. It is worth mentioning that for the low-cost ceramic membranes prepared by Vasanth et al. 2013 [53], Souza et al. 2021 [54], Emani et al. 2013 [18] and Suresh et al. 2016 [55] the estimated cost was found as 55–58, 97, 130 and 15–40 ($/m^2^), respectively.

## 4. Conclusions

New composite membranes (kaolin/sand and kaolin/zeolite) were successfully prepared by coating the kaolin layer over sand and a zeolite support. SEM images indicated that composite kaolin membranes have homogeneous surfaces and good distribution of kaolin particles. The optimized membranes kaolin/sand and kaolin/zeolite, sintered at 900 °C and 850 °C, show an average pore diameter in the range of 4–17 nm and 28 nm as well as water permeability of 491 L/h·m^2^·bar and 182 L/h·m^2^·bar, respectively. The treatment of the industrial electroplating wastewater exhibits a highly stabilized permeate flux in the range of 306–336 L/h·m^2^·bar using kaolin membranes at a temperature of 60 °C and optimal transmembrane pressure of 1 bar for kaolin/sand and 3 bar for kaolin/zeolite. In addition, good efficiency of the kaolin membranes in terms of turbidity and oil removal (>99%), COD rejection (>96%) and high metal removal (>94%) was achieved.

The good quality of these new composite kaolin membranes in the UF process for wastewater treatment was assessed by comparing their performances with respect to the nature of the support used. The highest flux of 336 L/h·m^2^ was determined for the kaolin/sand membrane at lower pressure of 1 bar, while a similar permeate flux of 306 L/h·m^2^ was achieved for the kaolin/zeolite membrane at relatively high pressure of 3 bar. Regarding the flux recovery ratio (FRR) and the flux decay ratio (FDR), the utilization of kaolin membrane over the sand support (kaolin/sand) for the wastewater treatment was more beneficial than the kaolin membrane coated on zeolite support (kaolin/zeolite). In addition, the fabrication cost of kaolin membranes was estimated at 17.88 $/m^2^ for kaolin/sand and at 20.92 $/m^2^ for kaolin/zeolite. Therefore, from these results, it can be concluded that good antifouling properties, high flux performances at lower transmembrane pressure and relatively low fabrication cost can be achieved with the composite kaolin membrane with sand support.

## Figures and Tables

**Figure 1 membranes-12-01066-f001:**
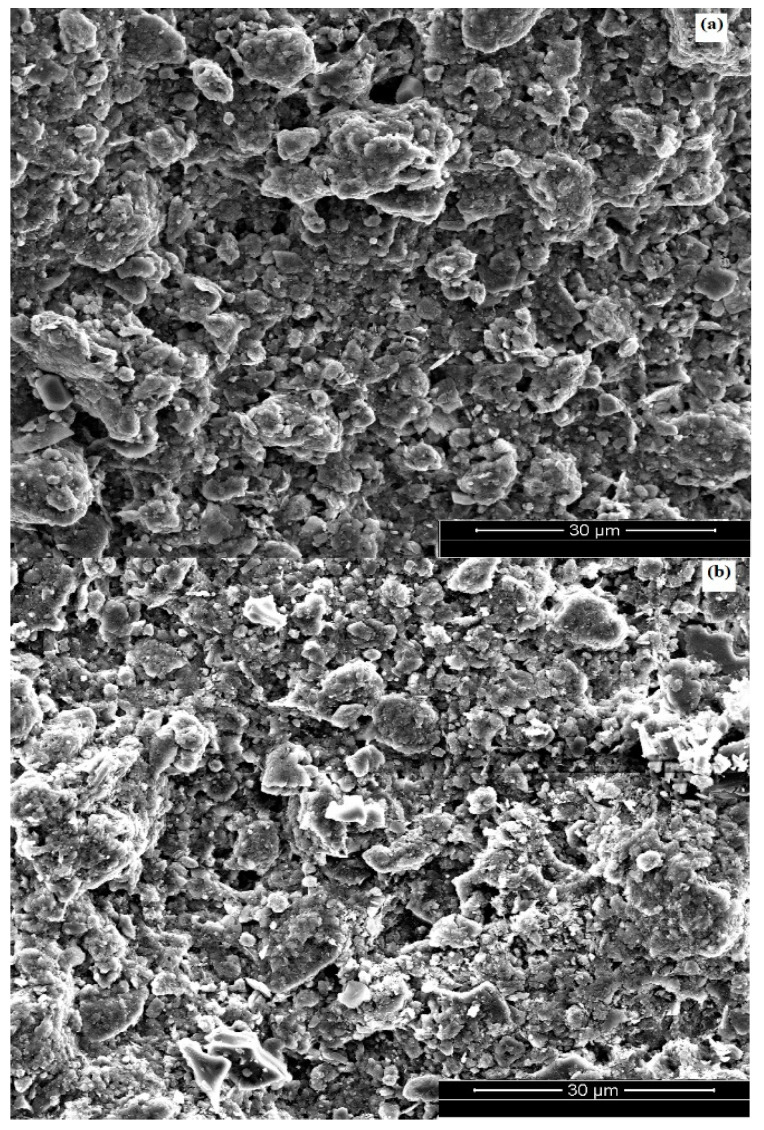
SEM micrographs of composite kaolin/sand membrane surface: sintered at 800 °C (**a**) 900 °C (**b**) and sintered at 1000 °C (**c**).

**Figure 2 membranes-12-01066-f002:**
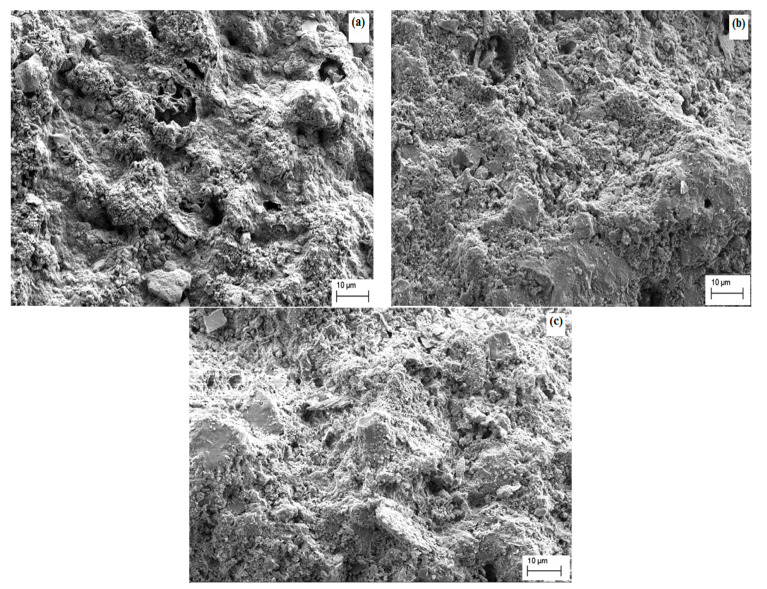
SEM micrographs of composite kaolin/zeolite membrane surface: sintered at 820 °C (**a**) 850 °C (**b**) and sintered at 880 °C (**c**).

**Figure 3 membranes-12-01066-f003:**
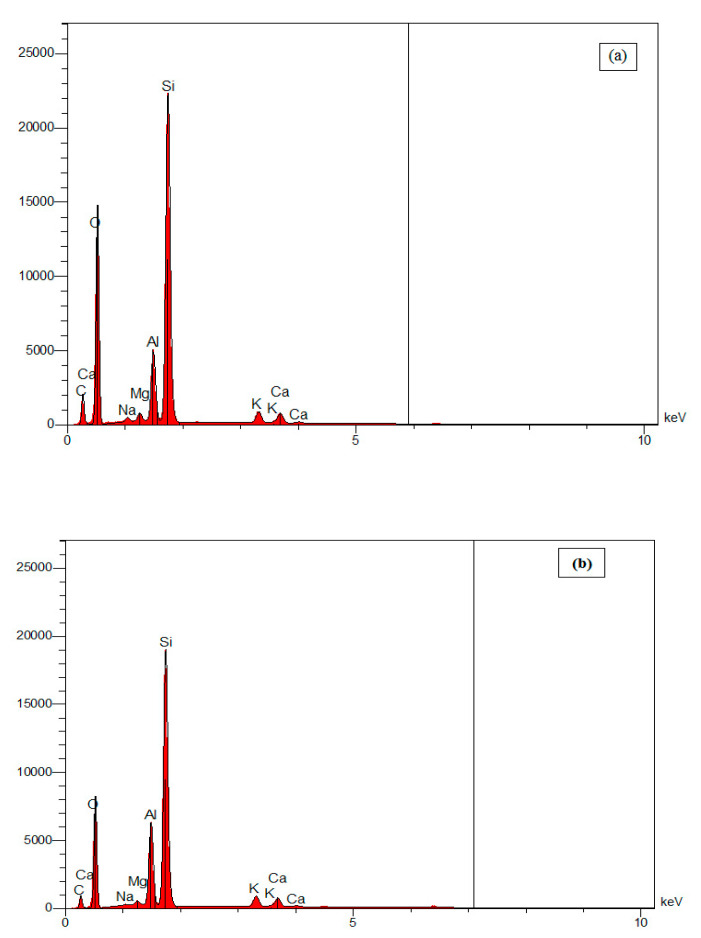
EDX spectra of the kaolin/sand (**a**) and kaolin/zeolite (**b**) membranes.

**Figure 4 membranes-12-01066-f004:**
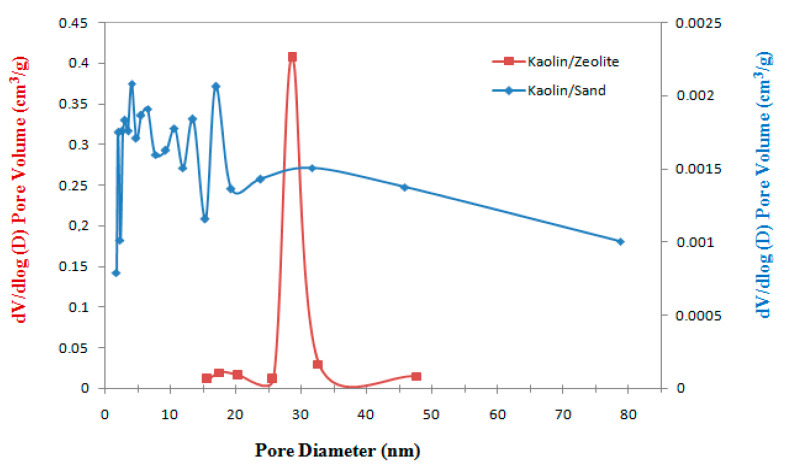
Distribution of pore diameters for kaolin/sand and kaolin/zeolite membranes.

**Figure 5 membranes-12-01066-f005:**
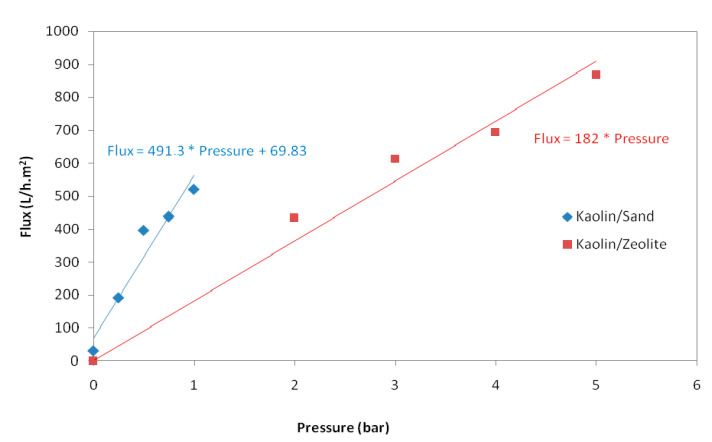
Evolution of water permeate flux of kaolin/sand and kaolin/zeolite membranes with pressure.

**Figure 6 membranes-12-01066-f006:**
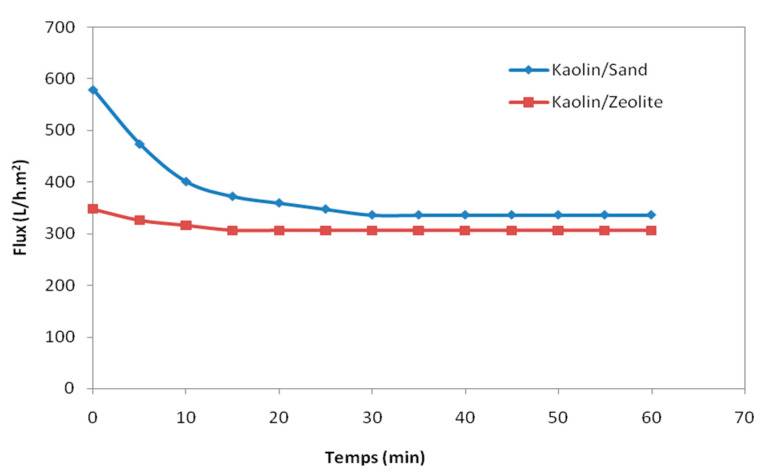
Permeate flux of kaolin/sand and kaolin/zeolite membranes as a function of time.

**Figure 7 membranes-12-01066-f007:**
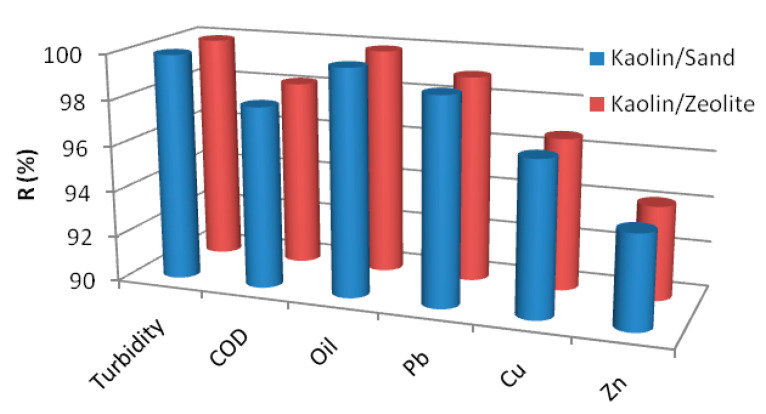
Retention of pollutants by different membranes at 1 bar for kaolin/sand and 3 bar for kaolin/zeolite.

**Figure 8 membranes-12-01066-f008:**
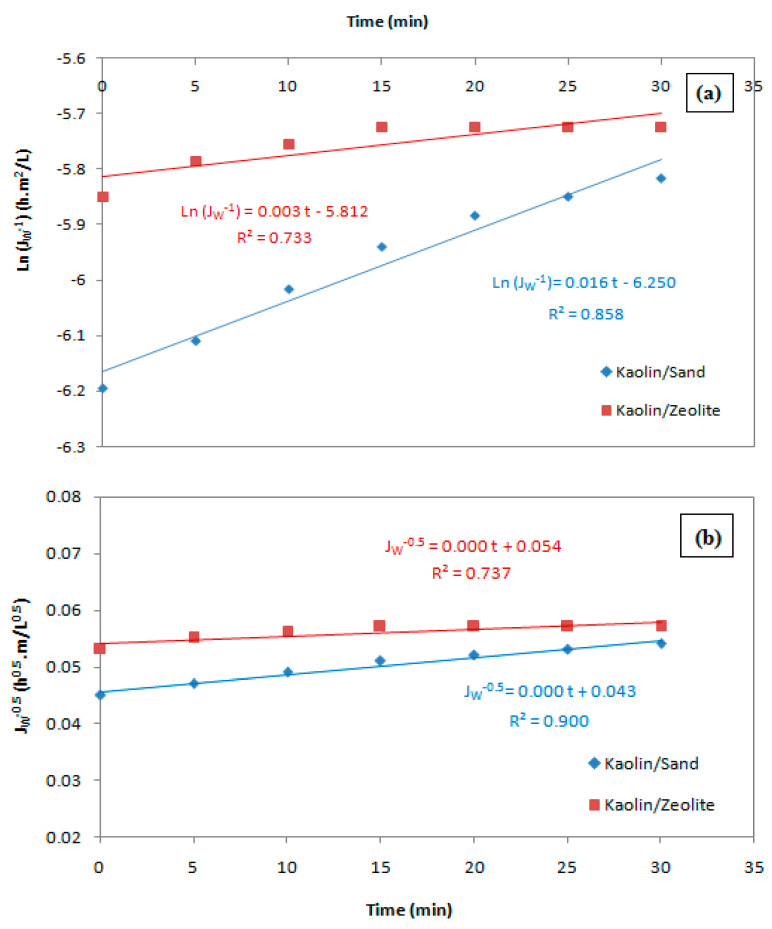
Linearized models of permeate fluxes of wastewater using kaolin membranes: complete pore blocking (**a**), standard pore blocking (**b**), intermediate pore blocking (**c**) and cake filtration (**d**).

**Figure 9 membranes-12-01066-f009:**
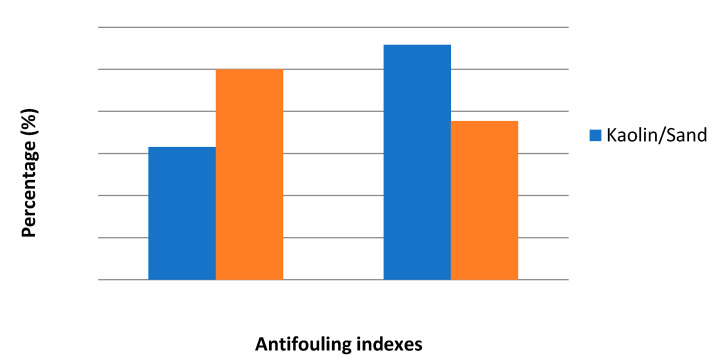
Antifouling parameters of kaolin membranes.

**Figure 10 membranes-12-01066-f010:**
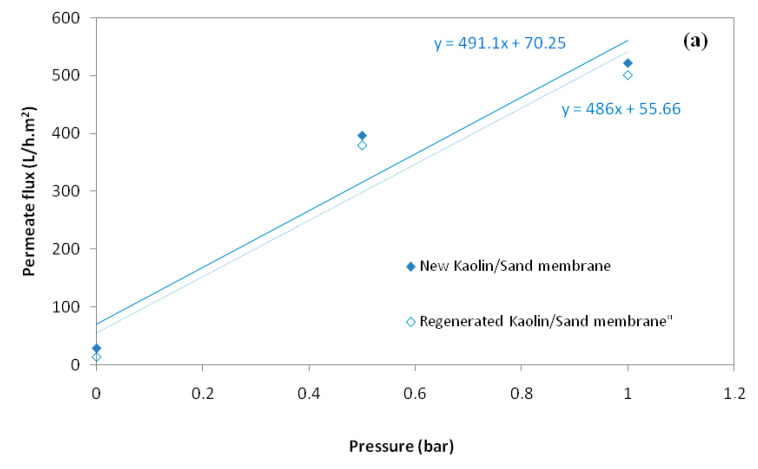
Water permeate flux versus transmembrane pressure for the new and regenerated membranes: kaolin/sand (**a**) and kaolin/zeolite (**b**).

**Table 1 membranes-12-01066-t001:** Characteristics of sand and zeolite supports.

Parameter	Sand Support	Zeolite Support
Sintering temperature (°C)	1250	900
Pore size (µm)	10.36	0.55
Mechanical strength (MPa)	15.14	12.56
Water permeability (L/h·m^2^·bar)	3611	1218

**Table 2 membranes-12-01066-t002:** Main characteristics of the raw wastewater coming from the electroplating industry.

	pH	Conductivity (ms/cm)	Oil Content (mg/L)	COD (mg/L)	Turbidity (NTU)	Pb (mg/L)	Zn (mg/L)	Cu (mg/L)
Raw wastewater	6.94 ± 0.2	3.33 ± 0.4	65,500 ± 500	4950 ± 200	>6000	21.5 ± 0.02	10.4± 0.02	1.12 ± 0.02

**Table 3 membranes-12-01066-t003:** Parameters associated with various pore blocking models.

Blocking Model	Kaolin/Sand			Kaolin/Zeolite		
	K	J_0_	R^2^	K	J_0_	R^2^
Complete pore blocking	0.016	−6.25	0.858	0.003	−5.812	0.73
Standard pore blocking	0	0.043	0.9	0	0.054	0.737
Intermediate pore blocking	0.031	2.088 × 10^−3^	0.977	0.011	2.991 × 10^−3^	0.739
Cake filtration	0.157	4.354 × 10^−6^	0.989	0.072	8.952 × 10^−6^	0.747

**Table 4 membranes-12-01066-t004:** Details of cost estimation for fabrication of Kaolin/Sand and Kaolin/Zeolite membranes.

Cost of Raw Materials					
Raw Materials	Unit per Kg ($)	Amount of Raw Materials (g)		Cost ($)	
		Kaolin/Sand	Kaolin/Zeolite	Kaolin/Sand	Kaolin/Zeolite
Sand powder	0.029	336	-	0.0097	-
Zeolite powder	0.174	-	336	-	0.058
Kaolin powder	0.16	8	2	0.0012	0.0003
Distilled water	0.28	172	278	0.048	0.077
Porosity agents	1.29	64	64	0.082	0.082
Organic binder (PVA)	0.8	30	30	0.024	0.024
Total raw materials cost for fabrication of 15 membranes				0.165	0.241
Energy cost (Based on the power consumption)					
				Kaolin/Sand	Kaolin/zeolite
Mixer				0.031	0.031
Dry oven				0.027	0.027
Extruder				0.138	0.138
Furnace				0.086	0.086
Total production cost for fabrication of 15 membranes ($) (Surface of membrane = 1.7 × 10^−3^ m^2^)				0.447	0.523
Total production cost of membrane ($/m^2^)				17.88	20.92

## Data Availability

The data presented in this study are available on request from the corresponding author.

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
