# Peer review of "Development of Ultrafiltration Kaolin Membranes over Sand and Zeolite Supports for the Treatment of Electroplating Wastewater"

_membranes, 2022, doi:10.3390/membranes12111066_

Round 1

Reviewer 1 Report

Report on the Manuscript ID: membranes-1984572

Journal: Membranes

Title: Development of Ultrafiltration Kaolin Membranes over Sand and Zeolite Supports for the Treatment of Electroplating Wastewater

Authors: Hajer ALOULOU, Wala ALOULOU, Joelle DUPLAY, Lassaad BAKLOUTI *, Lasâad DAMMAK, Raja BEN AMAR

The presented paper involved development and characterization of new low-cost ultrafiltration (UF) membranes made from natural Tunisian kaolin. The asymmetric ceramic membranes were developed via Layer-by-Layer and slip-casting methods by direct coating on tubular supports previously prepared from sand and zeolite via the extrusion process. The overall data suggest that the developed kaolin membranes have the potential for remediation of oily industrial effluents contaminated by oil and heavy metals.

The title and the idea of the presented paper seems interesting to the readers of the journal membranes. After careful going through the manuscript it gives me good impression of its contents except some corrections and improvements are required.

1-    In introduction: page 2, 88-89, no need to mention the name of metal and its symbol; it is sufficient to mention symbol like Zn, Cr..etc.

2-    In experimental: page 3, line 118, what kind of residual particles that removed?; line 127 what authors mean by green membranes?; line 128, what kind of furnace be used please specify?; lines 134-135, what kind of instruments SEM, TEM and EDX please specify? Line 134, how authors know that their collected oily water samples are contaminated with heavy metals are they checked by analyses? What are these heavy metal cations? Specify? By what tools those authors measured their contaminants before and after treatment to apply to the used equations?? . Page 5, lines 188-189, the cleaning water permeability cannot judge efficiency of cleaning authors has to check and clarify??

3-    In results and discussion: Page 5, Fig 3 of EDX needs redrawing to become of an acceptable form? Page 5, line228, by what to authors measured pore diameters for the kaolin composite or calculate?? Pag8, line 276, How authors measured and calculate metals like Pb (99%), Cu (>96%) and Zn (94%)???? ; Figs 8 and 10, need redrawing to become suitable for publication in membrane journal???

Reviewer 2 Report

I think the manuscript can be accepted after minor revision. 1. From Fig. 4, it can be concluded that the pore diameters for the kaolin composite membranes are in the range of 4-17 nm for kaolin/sand and 26-34 nm for kaolin/zeolite. Obviously, the pore size distribution of Kaolin/sand membrane is very wide. Kaolin/zeolite membrane is better. I want to know how to get the range 4-17. 2.Generally, it is well known that the UF cannot be applied directly to remove heavy metals from wastewater due to its comparatively high pore size (2–50 nm). However, the separation of metal ions by the charged UF membranes can be explained by ion repulsion. Is that mean the surface of the kaolin membrane is positive charge? 3. The total production cost of membrane is only 17.88-20.92 ($/m2), much lower than the literature and commercial membrane. I want to know why the production cost is so low? Energy, raw material?

Reviewer 3 Report

This manuscript addresses the treatment of electroplating wastewater by ultrafiltration using two different ultrafiltration membranes such as kaolin/sand and kaolin/zeolite. Research outcomes are informative, and the paper could be further improved in the following aspects.

1.      In this article, the authors treated real wastewater to remove oil content. The authors need to specify how the calibration curve is prepared to know the actual oil concentration in real wastewater.

2.      From EDX analysis of sintered membranes, the peak for carbon was visible even after the membranes were sintered at high temperature. Please explain the source for carbon.

3.      As the separation of heavy metals is explained based on the surface charge of the membrane, it is advised to report the surface charge of the membrane for better understanding.

4.      The values in all Figures need to be checked (“,” was used in place of “.”)

5.      Authors should check the pure water flux data of kaolin/sand membrane. At zero transmembrane pressure, flux was reported to be 69.83 L/h.m2 (Figure 5). Please check.

6.      In equation 1, Rx represents rejection of the membrane. The word,“purification efficiency” (line 254, line 272) may be replaced with “rejection” in order to avoid confusion.

7.      The dimension of the membrane needs to be provided in section 3.3.

8.      In Figure 6, the flux reduction was lower for kaolin/zeolite membrane, though the transmembrane pressure was higher (3 bar) than kaolin/sand membrane (1 bar). Please provide a possible reason.

9.      From FRR values, it appears that fouling is more severe for kaolin/zeolite membrane. However, it was mentioned in section 3.2.1, “therefore, it seems that the fouling is more important for the kaolin/sand membrane” (line 263-264). Authors should justify this.
